# Proteomics of the Honeydew from the Brown Planthopper and Green Rice Leafhopper Reveal They Are Rich in Proteins from Insects, Rice Plant and Bacteria

**DOI:** 10.3390/insects11090582

**Published:** 2020-09-01

**Authors:** Jinghua Zhu, Kunmiao Zhu, Liang Li, Zengxin Li, Weiwei Qin, Yoonseong Park, Yueping He

**Affiliations:** 1Hubei Insect Resources Utilization and Sustainable Pest Management Key Laboratory, College of Plant Science and Technology, Huazhong Agricultural University, Wuhan 430070, China; zhujinghua624@163.com (J.Z.); kunmiaozhu@webmail.hzau.edu.cn (K.Z.); liliang199492@163.com (L.L.); duobbi@163.com (Z.L.); 15927010047@163.com (W.Q.); 2Department of Entomology, Kansas State University, Manhattan, KS 66506, USA; ypark@ksu.edu

**Keywords:** honeydew, *Nilaparvata lugens*, *Nephotettix cincticeps*, plant immune systems, protein, proteome, shotgun LC–MS/MS

## Abstract

**Simple Summary:**

Plant sap-sucking insects can secrete saliva into the plant and excrete honeydew on the plant surface, which both interfere with the plant’s immune system. However, knowledge about the composition of insect honeydew and the roles it plays in plant defenses is limited. In the present study, a diversity of proteins in the honeydew of two major rice pests, the brown planthopper and green rice leafhopper, were identified by way of proteomic analysis. Results revealed that the honeydew contains some insect proteins that originate from the saliva and guts, proteins that originate from plant sap via the insect digestion system as well as many bacterial proteins from insect or plant symbionts and from plant surfaces. These abundant proteins from insects, microbes and plants in the honeydew might be elicitors, effectors or self-recognized molecules for plant defenses, which provide further insights into their roles in the multitrophic interactions of plants–insects–microbes.

**Abstract:**

Honeydew is a watery fluid excreted by plant sap-feeding insects. It is a waste product for the insect hosts. However, it plays important roles for other organisms, such as serving as a nutritional source for beneficial insects and bacteria, as well as elicitors and effectors modulating plant responses. In this study, shotgun LC–MS/MS analyses were used to identify the proteins in the honeydew from two important rice hemipteran pests, the brown planthopper (*Nilaparvata lugens*, BPH) and green rice leafhopper (*Nephotettix cincticeps*, GRH). A total of 277 and 210 proteins annotated to insect proteins were identified in the BPH and GRH honeydews, respectively. These included saliva proteins that may have similar functions as the saliva proteins, such as calcium-binding proteins and apolipophorin, involved in rice plant defenses. Additionally, a total of 52 and 32 *Oryza* proteins were identified in the BPH and GRH honeydews, respectively, some of which are involved in the plant immune system, such as Pathogen-Related Protein 10, ascorbate peroxidase, thioredoxin and glutaredoxin. Coincidently, 570 and 494 bacteria proteins were identified from the BPH and GRH honeydews, respectively, which included several well-known proteins involved in the plant immune system: elongation factor Tu, flagellin, GroEL and cold-shock proteins. The results of our study indicate that the insect honeydew is a complex fluid cocktail that contains abundant proteins from insects, plants and microbes, which may be involved in the multitrophic interactions of plants–insects–microbes.

## 1. Introduction

The rice brown planthopper (BPH), *Nilaparvata lugens* (Stål), and green rice leafhopper (GRH), *Nephotettix cincticeps* (Uhler), are two of the most economically important insect pests affecting rice products in Asia [1]. These insects cause heavy losses to rice yields by directly sucking plant sap or transmitting plant viruses. Rice hoppers suck fluid from the vascular bundle and excrete the liquid honeydew on the surface of rice plants. The honeydew of rice hoppers is primarily considered to contain only sugars, amino acids and other chemicals; therefore, it is often viewed as a kairomone and also a nutrient source to predators, such as the mirid bugs *Cyrtohinus lividipennis* and *Tytthus parviceps* [2]. BPH honeydew was recently reported to elicit direct defense (phytoalexins accumulation) and indirect defense (release of volatile organic compounds to attract natural enemies of herbivores) responses in the rice plant via honeydew-associated microbes [3]. However, the elicitors or effectors in honeydew potentially involved in rice plant defenses have not been determined to date.

Plants attacked by herbivores and pathogens evolve an equally diverse set of immune responses [4,5]. Plant defenses are triggered by either herbivore-associated molecular patterns (HAMPs) or microbial (or pathogen)-associated molecular patterns (MAMPs or PAMPs) [6,7,8,9,10,11,12,13]. Insects and microbes also produce a suite of effectors to modulate plant defenses. In general, HAMPs and herbivore-associated effectors occur in the oral secretions, saliva, oviposition fluids and digestive waste products (e.g., frass and honeydew) of insect herbivores and herbivore-associated endosymbionts [14]. Honeydew excreted from the insect anus consists of waste products of the insect’s metabolism eliminated via the gut, as well as the residue of the ingested plant sap after digestion and assimilation in the insect gut [15]. It also contains symbionts and external environment microorganisms [3,16,17]. When deposited onto the plant or other surfaces, it serves as an excellent growth medium for microorganisms. Honeydew contains a variety of chemicals and proteins from different organisms, such as insects and their symbionts; plants, and the microorganisms they harbor; and from microorganisms on the plant surface. It may therefore contain both elicitors (HAMPs, MAMPs) and effectors to elicit or modulate plant responses. However, knowledge about the composition of insect honeydew and the roles it plays in plant defenses is limited. To date, there has been only one publication describing a proteomic analysis of insect honeydew from the pea aphid (*Acyrthosiphon pisum*). The authors observed that the honeydew contained many insect proteins and microbial proteins, such as chaperonin, GroEL and Dnak chaperones, elongation factor Tu (EF-Tu), and flagellin, which might be involved in plant–aphid interactions [18]. In the present study, shotgun LC–MS/MS analysis was used to identify the proteins present in the BPH and GRH honeydews, which originated from the insects, rice plant and bacteria. The results of this study could help to elucidate the role of honeydew in plant–insect interactions.

## 2. Materials and Methods

### 2.1. Plants and Insects

Rice plants (*Oryza sativa* L.) of the Taichung Native 1 (TN1) variety were used for insect rearing and honeydew collection. The planthoppers and leafhoppers were collected in 2013 from rice fields on the Huazhong Agricultural University campus, Wuhan, China, and were reared continuously on TN1 rice seedlings in the laboratory at 27 ± 1 °C, 80 ± 10% humidity and a 14:10 h (light:dark) photoperiod. Insects were frequently collected from the field to enhance the genetic diversity.

### 2.2. Honeydew Collection

The parafilm sachet method was used to collect the honeydew [19]. Before honeydew collection, parafilm square pieces were bathed in 70% ethanol solutions, kept dry and then used for preparing the parafilm sachets (4 cm × 4 cm). The stems of the rice plants in the tillering stage were wiped gently using a cotton ball with 70% ethanol. Then an adult (2–3 days after eclosion) was enclosed in a parafilm sachet and fixed on a rice stem. After 24 h, honeydew droplets within the sachet were collected using a 10 μL pipette tip and transferred into sterile microcentrifuge tubes.

### 2.3. Protein Digestion

In total, approximately 500 μL of honeydew excreted from 30 adults was collected. The solution was incubated at 60 °C for 1 h after adding 0.05 M trichloroethyl phosphate (TCEP, Sigma, Darmstadt, Germany). The reaction solution was alkylated by adding a 55 mM methyl-methyl-thiomethyl sulfoxide (MMTS, Sigma) solution at room temperature in the dark for 45 min. The sample was concentrated using a centrifugal device (Pall Nanosep^®^, MWCO 10 kDa, Port Washington, NY, USA) and centrifuged at 12,000× *g* for 20 min. Then the sample was spun down twice after adding 100 μL of a urea solution (8 M urea, pH = 8.5) and spun down three times after adding 100 μL of 0.25 M triethylammonium bicarbonate (TEAB, Sigma). The protein suspension was digested with 50 μL of 0.5 M TEAB and 2 μL of 2% trypsin (Promega) and incubated at 37 °C overnight. Two microliters of 1% trypsin was added again and held at 37 °C for 4 h. Then, the sample was transferred to a new tube and spun down. The elution solution was collected and dried at a low temperature in vacuum.

### 2.4. LC–MS/MS Analysis

Before MS identification, peptides were dissolved using a solution of 0.1% formic acid and 2% acetonitrile [20], and the supernatant was collected after centrifuging at 13,200 rpm at 4 °C for 10 min.

LC−MS/MS was performed on a Thermo Dionex Ultimate 3000 RSLCnano system (Thermo Scientific, Waltham, MA, USA) coupled to a Thermo Scientific Q Exactive mass spectrometer (Thermo Scientific). Peptides were trapped on C18 columns (300 μm idx 5 mm, Acclaim PepMap RSLC C18, 5 μm, 100 Å, Thermo) and separated on an analytical column (Acclaim PepMap 75 μm X 150 mm, C18, 3 μm, 100 Å, Thermo). Mobile phase A was 0.1% formic acid, and mobile phase B was 0.1% formic acid and 80% acetonitrile. The gradient was run as follows: 5% B to 90% B at 65 min at a flow rate of 300 nL/min.

Separated MS data were analyzed using a Thermo Scientific Q Exactive mass spectrometer. The first single ion monitoring method employed an *m/z* 350–1800 mass selection, an orbitrap resolution of 70,000, target automatic gain control (AGC) values of 3 × 10^6^ and maximum fill times of 40 ms. The second single ion monitoring method employed a resolution of 17,500, target AGC values of 1 × 10^5^, maximum individual fill times of 60 ms and a normalized collision energy of 27.

### 2.5. Protein Identification

Raw data from the LC–MS/MS analyses were searched using MASCOT software (version 2.2.04, Matrix Science, Boston, MA, USA). To identify the insect proteins, each MS/MS peptide of the BPH honeydew sample was searched against the UniProt protein database (*Nilaparvata lugens*, http://www.UniProt.org/taxonomy/108931, 1872 proteins), a custom *N. lugens* whole body transcriptome database (NlWB, 16,440 predicted proteins, Appendix A in Liu et al. [21]) and a custom *N. lugens* salivary gland transcriptome database (NlSG, 14,203 predicted protein sequences, Appendix A in Liu et al. [21]). Each MS/MS peptide of the GRH honeydew sample was searched against a custom re-assembled GRH whole body transcriptome database (NcWB, 17,181 predicted protein sequences) based on an NCBI SRA data (accession number SRX4777341, GRH adult whole body; RNA-Seq using Illumina HiSeq 1500; https://www.ncbi.nlm.nih.gov/sra/SRX4777341) and a custom re-assembled GRH salivary gland transcriptome database (NcSG, 16,123 predicted protein sequences) based on an NCBI SRA data (accession number DRX016686, GRH female salivary gland; RNA-Seq using Illumina HiSeq 2000; https://www.ncbi.nlm.nih.gov/sra/DRX016686) using Trinity software (http://trinityrnaseq.sourceforge.net/). All assembled sequences were annotated against the NCBI protein non-redundant (NR), String and KEGG databases through BLASTX with a cut-off expected value (E-value) of 10^−5^. To identify the rice plant and bacterial proteins, the LC–MS/MS raw data were searched against the UniProt Knowledgebase Swiss-Prot database (*Oryza*, 241,573 protein sequences; Bacteria, 323,233 protein sequences).

The following options for protein identification were used: (i) fixed modification, carbamidomethyl (C); (ii) variable modification, oxidation (M); (iii) enzyme, trypsin; (iv) maximum missed cleavage, 2; (v) peptide mass tolerance, 20 ppm; (vi) fragment mass tolerance, 0.6 Da; (vii) mass values: monoisotopic; and (viii) significance threshold: 0.05. The ions score was −10*Log (P), where P is the probability that the observed match is a random event. The threshold ions scores (*P* < 0.05) were suggested by MASCOT for confident single peptide identifications in different databases. The matched overlapping proteins identified from different BPH and GRH databases were deleted after blasting. High confidence proteins were identified with at least two unique matched peptides.

BLAST alignment was conducted with a cutoff E-value of <10^−5^ and identities >40% using BioEdit (http://www.mbio.ncsu.edu/BioEdit/bioedit.html). The proteins with identities of 40~50% were further verified by a BLAST search against the NCBI database.

## 3. Results

### 3.1. Identification of Insect Proteins in the BPH Honeydew

A total of 281 proteins from the BPH honeydew were identified by shotgun LC–MS/MS based on mass spectrometry data searched against the *N. lugens* Knowledgebase (UniProtKB), NlWB and NlSG custom databases (Appendix A). Among these proteins, 246 proteins had at least two unique matched peptides (Table 1; peptide sequences are listed in Appendix A). A total of 98.6% (277/281) of the proteins were annotated to the insect proteins, primarily from BPH (91.1%: 256/281).

After comparing 277 insect proteins as identified in the BPH honeydew to our custom BPH transcriptome databases of salivary glands and intestines (midgut, hindgut and Malpighian tubules, He et al. unpublished data), 80.5% (223/277) and 94.2% (261/277) proteins obtained hits for the BPH salivary gland transcripts and intestine transcripts, respectively (Appendix A). Nine proteins were found only in the salivary gland and not in the intestine, while 47 proteins were found only in the intestine and not in the salivary gland. The presence of salivary gland-specific proteins in the BPH honeydew indicated the possibility that these proteins are first secreted into rice, then ingested into insect guts and then excreted. The intestine-specific proteins in the BPH honeydew contained a few enzymes involved in detoxification or digestion, such as P450s, carboxylesterase, thioredoxin and serine proteinase.

The insect proteins identified in the BPH honeydew were grouped into putative functional categories (Table 1), including (1) enzymes, such as oxidoreductases, hydrolases, phosphatases, peptidases, kinases, transferases, lyases and ligases; (2) ATPases; (3) calcium-binding protein; (4) cytoskeleton-associated proteins, such as actin filaments, intermediate filaments (IF) proteins and microtubules; (5) genetic information processing; (6) heat-shock proteins; (7) other non-enzyme proteins, such as sugar transporter, lipophorin, channels and receptors; and (8) unknown proteins.

### 3.2. Identification of Insect Proteins in the GRH Honeydew

A total of 269 proteins from the GRH honeydew were identified by shotgun LC–MS/MS based on mass spectrometry data searched against the custom NcWB and NcSG transcriptome databases (Appendix A). Among them, 79 proteins were identified from both the NcWB and NcSG transcriptomes, 53 proteins were only found in NcSG and 137 proteins were only found in NcWB. Among these proteins, 240 proteins had at least two unique matched peptides (Table 2; peptide sequences are listed in Appendix A). A total of 78.1% (210/269) of the proteins were annotated to insect proteins. Only one protein (NcSP19) was annotated to known GRH proteins.

The insect proteins identified in the GRH honeydew were grouped into putative functional categories (Table 2), including (1) enzymes, such as oxidoreductases, hydrolases, phosphatases, peptidases, kinases, transferases, lyases and ligases; (2) ATPases; (3) calcium-binding proteins; (4) cytoskeleton-associated proteins, such as actin filaments, intermediate filaments (IF) proteins and microtubules; (5) genetic information processing; (6) heat-shock proteins; (7) transporters; and (8) others and unknown proteins.

Comparison of the insect proteins in the BPH and GRH honeydews yielded a total of 21 orthologs: glyceraldehyde-3-phosphate dehydrogenase, ATP synthase subunit alpha, ATP synthase-beta, histone H2B, actin, beta-enolase, vacuolar ATP synthase subunit E, myosin-1, HSP70, HSPA8, succinate semialdehyde dehydrogenase, aspartyl-tRNA synthetase, succinyl-CoA ligase subunit alpha, nuclear hormone receptor FTZ-F1 beta, protein strawberry notch-like, keratin II, HSP100 clpX-like, ABCG1, ATP-dependent zinc metalloprotease YME1, cytoplasmic aconitate hydratase-like, ubiquitin/ribosomal protein S27Ae fusion protein and tropomyosin 2.

### 3.3. Identification of Rice Proteins in the BPH Honeydew

By searching the *Oryza* protein Knowledgebase (UniProtKB), a total of 52 proteins were identified from the LC-MS/MS data of the BPH honeydew (Appendix A; the peptide sequences are listed in Appendix A). Among these proteins, nine had at least two unique matched peptides (Table 3), which includes six oxidoreductases (glutaredoxin-C6, thioredoxin H1, thioredoxin, L-ascorbate peroxidase 1, glutathione S-transferase DHAR2, Cu/Zn superoxide dismutase), pathogen-related protein 10 (PR10), putative acyl-CoA-binding protein RPP10, and heat-shock 70 kDa protein (HSP70).

### 3.4. Identification of Rice Proteins in the GRH Honeydew

A total of 32 *Oryza* proteins were identified from the GRH honeydew (Appendix A; the peptide sequences are listed in Appendix A). Among them, four proteins (PR10, polyubiquitin 11, actin-2 and disease resistance RPP13-like protein) had at least two unique matched peptides (Table 4). PR10 and disease resistance RPP13-like protein are known plant pathogen-related proteins.

Comparison of rice plant proteins in the BPH and GRH honeydews yielded 11 proteins found in both honeydews: PR10, thioredoxin H1, glyceraldehyde-3-phosphate dehydrogenase 1, malate dehydrogenase, histone H4, ribulose bisphosphate carboxylase/oxygenase activase, elongation factor Tu, pentatricopeptide repeat-containing protein At5g12100, and 2-hydroxyisoflavanone dehydratase.

### 3.5. Identification of Bacterial Proteins in the BPH Honeydew

By searching the bacterial protein Knowledgebase (UniProtKB), a total of 570 proteins were identified from the LC–MS/MS data of the BPH honeydew (Appendix A; the peptide sequences are listed in Appendix A). Among these proteins, 491 had at least two unique matched peptides (Table 5), and 59.3% (338/570) of the proteins were annotated to three major bacterial genera: *Acinetobacter* (29.6%, 169/570), *Escherichia* (15.8%, 90/570) and *Serratia* (13.9%, 79/570).

The identified bacterial proteins in the BPH honeydew were grouped into putative functional categories (Table 5), including (1) enzymes, such as oxidoreductases, hydrolases, phosphatases, peptidases, kinases, transferases, lyases, ligases, isomerases and translocase; (2) ATP synthases; (3) ion-binding proteins; (4) ABC transporters and other transporters; (5) heat-shock proteins; (6) genetic information processing; (7) other proteins, such as flagellin assembly proteins; and (8) unknown proteins.

### 3.6. Identification of Bacterial Proteins in the GRH Honeydew

A total of 494 bacterial proteins were identified from the GRH honeydew (Appendix A; the peptide sequences are listed in Appendix A). Among these proteins, 407 had at least two unique matched peptides (Table 6), and 51.0% (252/494) of the proteins were annotated to three major bacterial genera: *Acinetobacter* (30.6%, 151/494), *Escherichia* (14.0%, 69/494) and *Serratia* (6.5%, 32/494).

The identified bacterial proteins in the GRH honeydew were grouped into putative functional categories (Table 6), including (1) enzymes, such as oxidoreductases, hydrolases, phosphatases, peptidases, kinases, transferases, lyases, ligases, isomerases and translocases; (2) ATPases; (3) ion-binding proteins; (4) ABC transporters and other transporters; (5) heat-shock proteins; (6) genetic information processing; (7) other proteins, such as flagellin assembly proteins; and (8) unknown proteins.

## 4. Discussion

In the present study, using the shotgun LC–MS/MS analyses, more abundant proteins from insects and bacteria and some rice plant proteins were identified from the BPH and GRH honeydews, some of which have been reported to be involved in plant immune responses.

### 4.1. Insect Proteins in the BPH and GRH Honeydews May Be Involved in the Plant Immune System

Rice plant hopper saliva is secreted into the plant, which facilitates hopper feeding and interferes with the plant’s immune system [22,23,24]. In addition to saliva, honeydew also produces a suite of elicitors and effectors to modulate plant defenses. Compared with the identified saliva proteins of BPH [21,25,26], 21 proteins were identified in both saliva and honeydew proteomes of *N. lugens*: glyceraldehyde-3-phosphate dehydrogenase, carboxylesterase, venom dipeptidyl peptidase, ribosomal protein S6 kinase alpha-5-like, enolases, ATP synthases, EF-hand motif proteins, actin, keratin I, neurofilament heavy polypeptide-like, nuclear hormone receptor FTZ-F1 beta, phosphorylated adapter RNA export protein-like, ubiquitin-conjugating enzyme E2 J1-like, histone H2B, DNA mismatch repair ATPase MutL, COX assembly mitochondrial protein, maspardin-like and heat-shock protein 70. Compared with the identified GRH saliva proteins [27], five proteins were identified in both saliva and honeydew proteomes of GRH: glyceraldehyde-3-phosphate dehydrogenase, aminopeptidase N-like, beta-enolase, polyubiquitin-B and heat-shock protein 70.

Currently, only a few saliva proteins of BPH and GRH have been reported to be involved in the rice plant immune system: EF-hand motif proteins (NlSEF1 and NcSP84) [24,28]), salivary sheath proteins (NlShp and Nlsalivap-3) [25,29], a mucin-like protein (NlMul) [30], endo-β-1,4-glucanase (NlEG1) [23], a saliva-specific laccase (NcLac1S) [31,32], GRH β-glucosidase [33], a salivary GRH protein NcSP75 [34], protein disulfide-isomerase (Nl12), apolipophorin III (Nl16), cysteine-rich protein (Nl28) and *N. lugens*-specific salivary protein (Nl43) [35]. Among these, apolipophorin III was found in the BPH honeydew. Apolipophorin III is known to play an important role in lipid transport and innate immunity in insects [36]. Apolipophorins secreted into plants via insect saliva may interact with plant defense signaling compounds, lipids and free fatty acids, as well as play an important role in plant–insect interactions. It was reported that apolipophorin III (Nl16), a salivary protein of BPH, induced cell death and elicited SA-related marker genes PR1 and PR2 in rice [35]. This protein was also found in the BPH honeydew (A0A1I9WL93), which suggests that the BPH honeydew may be an elicitor of defense responses in rice plants.

Calcium-binding proteins in the saliva of aphids and other phloem-ingestion insects play a key role in counteracting the sieve-tube occlusion defenses in plants [37,38]. BPH feeding can induce callose deposition in the sieve-tube of rice [39]. A knockdown of the BPH salivary EF-hand calcium-binding protein (GenBank: AOM63273.1) caused the elicitation of higher levels of Ca^2+^ and H_2_O_2_ in rice, suggesting that it functions as an effector for defense responses in rice [24]. None of the three EF-hand proteins identified in the BPH honeydew was the same as the EF-hand protein (GenBank: AOM63273.1) as reported in Ye et al. [24]. A GRH salivary EF-hand calcium-binding protein (NcSP84) was detected in the phloem sap of rice exposed to leafhoppers and may function as an effector for defense responses in rice [28]. Although NcSP84 was not found in the GRH honeydew, seven calcium-binding proteins were, however, detected. Whether these calcium-binding proteins and other salivary proteins in the BPH and GRH honeydews have any function in plant–insect interactions need to be further studied.

### 4.2. Plant Proteins in the BPH and GRH Honeydews Involved in the Plant Immune System

When plant proteins from the insect gut are deposited on leaf surfaces, plants are assumed to induce “self-recognition” as a means for a defense mechanism. “Self-recognition” is a phenomenon in plants in which plant defenses are activated by recognizing their own molecules or metabolized compounds transferred via the ingestion and secretion or excretion system of herbivores [40]. For example, *A. pisum* honeydew contains plant-derived SA that suppresses herbivore-induced defenses in plants [41]. Whitefly honeydew can modulate plant SA signaling by glycosylation of plant SA to SA glycoside [42]. Plant chitinases from *Spodoptera frugiperda* larval frass suppress herbivore defenses in maize (*Zea mays*) [43].

Among the rice proteins identified in the BPH honeydew, six are oxidoreductases involved in the plant reactive oxygen species (ROS) signaling: L-ascorbate peroxidase 1, thioredoxin H1, thioredoxin H2, glutaredoxin-C6, glutathione S-transferase dehydroascorbate reductase 2 (DHAR2) and Cu/Zn superoxide dismutase. Biotic stresses, such as insect herbivory, change the metabolic machinery of the plant cell, leading to the production of reactive oxygen species (ROS) [44]. Plants have developed several enzymatic systems to protect against oxidative damage caused by these ROS, including superoxide dismutases (SOD), catalases, ascorbate peroxidases, thioredoxins, glutaredoxins and glutathione S-transferases [45,46]. In addition to maintaining the steady-state level of ROS, these enzymes are also involved in the signaling cascades triggered in response to insect herbivory or pathogen attack [44].

Ascorbate peroxidase is a key antioxidant enzyme for metabolizing stress-provoked ROS [47]. Cytosolic ascorbate peroxidase activity in soybean leaves was increased after *Aphis craccivora* infestation [47]. Rice ascorbate peroxidases were upregulated upon wounding (by cut) and blast pathogen (*Magnaporthe grisea*) attack [48]. Thioredoxin H is one of the major proteins in rice phloem sap [49]. Thioredoxin H was differentially expressed in the rice sheath infected with the symbiotic bacterium, *Pseudomonas fluorescens* [50]. BPH feeding induced a significant change in the abundance of thioredoxin H in the phloem sap of rice [51]. Glutaredoxins (GRXs) are ubiquitous oxidoreductases that play a crucial role in response to oxidative stress by reducing disulfides in the presence of glutathione (GSH) [52]. Overexpression of the rice and *Arabidopsis* CC-type GRXs increased susceptibility to infection by the necrotrophic pathogen *Botrytis cinerea* and elevated endogenous hydrogen peroxide (H_2_O_2_) levels [53]. Glutathione S-transferase (GST) is a glutathione-dependent detoxifying enzyme that plays critical roles in stress tolerance and detoxification metabolism in plants. Plant GSTs can be divided into 6 categories: Phi, Tau, Zeta, Lambda, Theta and Dehydroascorbate reductases (DHAR) [54]. Glutathione-S-transferases can detoxify radicals [55], and their expressions are significantly upregulated by aphid feeding in systemic phloem tissue [56]. DHAR in rice plays roles as a plant GST during the bacterial leaf blight pathogen (*Xanthomonas oryzae* pv. *Oryzae*, Xoo) infection along with its usual antioxidative function [57]. SOD is the most important enzyme that responds to ROS, dismutating superoxide to H_2_O_2_ [58]. Cu/Zn superoxide dismutase (Cu/Zn SOD)-overexpressing *Arabidopsis thaliana* showed enhanced resistance to saline-sodic stress and growth tolerance to H_2_O_2_ stress [59,60]. The expression level of Cu/Zn-SOD in rice was induced by an elicitor from the fungus *Magnaporthe oryzae*, CSB I [61].

Additionally, some *Oryza* proteins, such as PR-10, disease resistance RPP13-like protein, acyl-CoA-binding protein RPP10 and HSP70, identified in the BPH honeydew, are known to be involved in the response to insect herbivory or pathogen attack. PR-10 is present in many plants and has been reported to possess ribonuclease activity and a role in defense against plant pathogens [62]. In rice, PR10 can be induced by MSP1 (*M. oryzae* snodprot1 homolog), an elicitor from *M. oryzae* that triggers the defense responses in rice [63]. A plant disease resistance protein, Recognition of *Peronospora Parasitica* 13-like (RPP13-like), belongs to the nucleotide-binding site and leucine-rich repeat (NBS-LRR) family [64]. NBS-LRR has been suggested to be the largest class of known plant disease resistance proteins (R proteins) that can either directly or indirectly recognize the presence of pathogens [64]. RPP13-like genes play important roles in the resistance of various plant diseases [65]. Acyl-CoA-binding proteins (ACBPs) are thought to facilitate the intracellular transport of fatty acids and lipids, which are well known regulators of plant defense [66]. ACBPs in *A. thaliana* are required for cuticle development as well as defense against microbial pathogens [67]. In this study, one putative ACBP protein, RPP10, identified in the BPH honeydew, was reported to be one of the major proteins in rice phloem sap [68]. HSP70s are a class of ubiquitous and highly conserved proteins that are fundamental in developmental processes and function in various abiotic stress-responses in plants [69]. In addition, HSP70 was reported to be induced upon infection with plant pathogens [63,70]. These rice proteins related to plant defenses were detected in the BPH and GRH honeydews, which suggests that the honeydew of rice hoppers could play a role in weakening the plant defense for insects or pathogens attack. Thioredoxin H and RPP10, two of major phloem proteins of rice plant, were detected also in the BPH and GRH honeydews, which suggests that some of the rice proteins could be ingested into the midgut of rice hoppers and excreted onto the surface of rice plants.

### 4.3. Bacterial Proteins in the BPH and GRH Honeydews Involved in the Plant Immune System

Abundant bacterial proteins were detected in the BPH and GRH honeydews, and some of them have been reported to be involved in the plant immune system: elongation factor Tu (EF-Tu), flagellin, GroEL (hsp60, chaperonin), cold-shock proteins (CSPs) and superoxide dismutase (SOD) [71,72,73,74]. Additionally, a high proportion of proteins involved in genetic information processing were also found, but their potential roles in the plant or insect immune system have not been elucidated.

EF-Tu, one of the most abundant and conserved proteins in bacteria, also acts as a bacterial pathogen-associated molecular pattern (PAMP) in *Arabidopsis* and other *Brassicaceae* species [71]. An EF-Tu receptor (EFR) found within *Brassica* lineages recognizes the highly conserved N-terminal 18 amino acids (elf18) in the native Ef-Tu molecule [71,75]. Because no EFR ortholog exists in rice, elf18 does not act as a PAMP in this species [75]. Rice can recognize the presence of EF-Tu by perception of a 50-amino-acid peptide of EF-Tu (EFa50) and mount immune responses by a currently unidentified receptor [76]. In the BPH honeydew, three EF-Tu proteins from *Acaryochloris*, *Acinetobacter* and *Kosmotoga* were identified. In the GRH honeydew, three EF-Tu proteins from *Acinetobacter*, *Paracoccus* and *Thiobacillus* were identified. Bacterial flagellin, the main building unit of the bacterial motility organ, is recognized as a PAMP in many different plant species, including rice [77]. The N terminal 22 amino acid long peptide (flg22) of flagellin from *Pseudomonas* species acts as a potent elicitor in *A. thaliana* and other plants [78]. The rice flg22 receptor (OsFLS2) can perceive flg22 derived from *Acidovorax avenae* (*P. avenae*), an important rice pathogen [79]. In this study, flagellin from *Serratia* and flagellar motor switch protein (FliG) from *Aquifex* were detected in the BPH honeydew. In the GRH honeydew, two flagellin proteins from *Serratia* and Rhizobium and four proteins related to the flagellin system from *Enterobacter*, *Bacillus*, *Aquifex* and *Pseudomonas* were identified. GroEL, from an obligate mutualist endosymbiotic bacterium *Buchnera aphidicola* in aphids, is a MAMP that induces defense responses in both tomato and *Arabidopsis* [72]. *Buchnera* GroEL has been identified in the saliva and honeydew of aphids [18,72,80]. In the BPH honeydew, three GroEL proteins from *Acinetobacter*, *Borrelia* and *Corynebacterium* were found. In the GRH honeydew, three GroEL proteins from *Acinetobacter*, *Azotobacter* and *Delftia* were detected. CSP was identified from bacteria as a PAMP in tobacco [73]. CSP was identified from *Xoo* as important for virulence against rice [81]. In the present study, two CSP proteins from *Escherichia* and *Buchnera* were identified in both BPH and GRH honeydews. Bacterial SOD, a major protein in the secretome of *X. campestris* pv. *campestris* and *E. coli*, was reported as a PAMP in tobacco [74]. In the BPH and GRH honeydews, three bacterial SOD proteins were identified, respectively.

In addition to the plant immune-related proteins, some insect proteins may be involved in the insect immune system (such as peptidoglycan recognition protein LC, β-1,3 glucan recognition protein, Toll-1 and reeler) and as detoxification enzymes (such as cytochrome P450s and carboxylesterases) [82,83], which can be indicators of the health of the insect.

In *A. pisum* honeydew, 60 insect proteins were identified using the 2D-PAGE technique [18]. After BLAST alignment, no ortholog of insect proteins were found in the BPH honeydew and *A. pisum* honeydews, while three orthologs (cathepsin B, katanin p60 ATPase-containing subunit A1 and maltase A3) were found in the GRH and *A. pisum* honeydews. Comparison of bacterial proteins in the BPH and GRH honeydews yielded 339 bacterial proteins found in both honeydews. In *A. pisum* honeydew, 33 bacterial proteins were identified [18]. Thirteen bacterial proteins were found in all of three honeydew proteomics of rice hoppers and *A. pisum*: glyceraldehyde-3-phosphate dehydrogenaseuridylate kinase, phosphoenolpyruvate carboxylase, phosphoserine aminotransferase, ATP synthase subunit beta, ATP-phosphoribosyltransferase, DNA-directed RNA polymerase subunit beta, Chaperone protein DnaJ, chaperone protein DnaK, chaperonin GroEL, elongation factor G, EF-Tu, aspartate--tRNA ligase and flagellin.

## 5. Conclusions

Our study provides further insights into hopper–rice plant interactions. The results in this study indicate that the honeydew from rice plant hoppers contains a diversity of proteins of different origins, some of which might be elicitors or effectors for plant defenses, indicating that honeydew can play a notable role in hopper–rice plant interactions. Insect honeydew is a fluid cocktail. The honeydew contains some insect proteins that originate from the saliva, which might play similar roles (HAMPs or effectors) to that of saliva proteins for plant defenses. The honeydew also includes some plant proteins that originate from the residue of ingested plant sap via the insect digestion system, which might induce the self-recognized immune responses of the plant. Insect honeydew also harbors many bacterial proteins from insect or plant symbionts and from plant surfaces, some of which could be MAMPs or HAMPs that alter plant defense responses to herbivores and pathogens. A mixture of putative plant immune-related proteins or other molecules in insect honeydew might have a combined action (synergism or inhibition) on plant immune responses. Insect honeydew with abundant proteins from different origins cast a new light on understanding the multitrophic interaction of plants–insects–microbes. However, further research is warranted to determine their quantities in the honeydew, and their specific functions in their interactions.

## Figures and Tables

**Table 1 insects-11-00582-t001:** Identified proteins from the brown planthopper (BPH) honeydew against the BPH protein databases (matched peptides > 1).

Classification	No.	Protein Names
Oxidoreductase	21	CYP417B1, CYP4G76, CYP4C78, CYP417A1, CYP314A1, CYP3A8-like, CYP3A25, CYP6CS1, thioredoxin, multicopper oxidase 1, kynurenine 3-monooxygenase, glyceraldehyde-3-phosphate dehydrogenase, malate dehydrogenase, saccharopine dehydrogenase, homoisocitrate dehydrogenase, NADH dehydrogenase subunit 6, dehydrogenase/reductase SDR 4-like, glutamate dehydrogenase, HIF hydroxylase
Hydrolase	9	carboxylesterase, easter-4, β-1,3 glucan recognition protein-2, chitinase-like protein EN03, beta-N-acetylhexosaminidase, peptide-N(4)-(N-acetyl-beta-glucosaminyl) asparagine amidase, juvenile hormone epoxide hydrolase, alakaline phosphatase 2
Peptidase	5	ATP-dependent zinc metalloprotease YME1, serine proteinase stubble-like, venom dipeptidyl peptidase, gamma-glutamyltranspeptidase 1-like, asparagine synthetase
Kinase	8	ribosomal protein S6 kinase alpha-5-like, ATP-dependent 6-phosphofructokinase, serine/threonine-protein kinase tousled-like 2, serine/threonine-protein kinase 26, myristoylated alanine-rich C-kinase substrate-like, 5′-AMP-activated protein kinase catalytic subunit alpha-2, adenosine kinase-like, tyrosine-protein kinase shark
Transferase	13	enolase 2, enolase, aspartate aminotransferase, tRNA (cytosine-5-)-methyltransferase, protein arginine methyltransferase NDUFAF7, phosphatidate cytidylyltransferase, peroxisomal carnitine O-octanoyltransferase-like, alpha-1,6-mannosyltransferase, UDP-N-acetylglucosamine pyrophosphorylase, glucose-6-phosphate isomerase 2, chitin synthase 1, fatty acid synthase, beta-1,4-glucuronyltransferase 1-like
Lyase	6	cystathionine beta-lyase, 2-oxoglutarate dehydrogenase, 4-hydroxy-2-oxoglutarate aldolase, glutamate decarboxylase, uridine 5′-monophosphate synthase
Ligase	4	adenylosuccinate synthetase, phosphoribosylformylglycinamidine synthase, succinate--CoA ligase, 4-coumarate--CoA ligase 1-like
ATPase	5	ATP synthase subunit alpha, ATP synthase subunit beta, ATP synthase subunit s, vacuolar ATP synthase subunit E, V-type proton ATPase subunit E
Calcium-binding protein	7	EF-hand motif protein, annexin, trichohyalin-like
Cytoskeleton proteins	20	alpha-actinin, actin, actin-like protein 6B, protein outspread, tropomyosin 2, troponin I, troponin T, myosin H, myosin-XVIIIa-like, moesin/ezrin/radixin homolog 1, myosin heavy chain, keratin I, keratin II, neurofilament heavy polypeptide-like; CAP-Gly domain-containing linker protein 2, microtubule-actin cross-linking factor 1, tubulin beta-1, tubulin-specific chaperone A
Genetic information processing	120	See Appendix A
Heat-shock protein	10	HSP70-2, HSP70-5, HSP70-3, HSP 70-6, HSP100, Hsc70, Hsc70-1, Mitochondrial-like 10 kDa heat-shock protein, Mitochondrial HSP75, T-complex protein 1 subunit gamma
Transporter	3	sugar transporter, apolipophorin-III, ATP-binding cassette sub-family G member 1-like protein
Membrane channel and receptor	5	voltage-dependent cation channel SC1, insulin receptor 1, insulin receptor 2, ryanodine receptor, ejaculatory bulb-specific protein 3-like
Unknown	16	interaptin-like, neuroendocrine protein 7B2, LOC111058029, LOC111063634, LOC111050340, LOC111054372, LOC111050355, LOC111043528, LOC111052895, LOC111050469, LOC107268379, LOC111060703

**Table 2 insects-11-00582-t002:** Identified insect proteins from the green rice leafhopper (GRH) honeydew (matched peptides > 1).

Classification	No.	Protein Names
Oxidoreductase	9	17-beta-hydroxysteroid dehydrogenase 13-like, CYP6a14, glucose dehydrogenase, glyceraldehyde-3-phosphate dehydrogenase, malate dehydrogenase, retinol dehydrogenase 11-like, succinate-semialdehyde dehydrogenase
Hydrolase	10	adenosylhomocysteinase, cAMP-specific phosphodiesterase 4, glucosidase 2 subunit beta, lysosomal alpha-glucosidase-like, kynurenine formamidase-like, maltase A1, maltase A3-like, phospholipase B-like 2, protein phosphatase PHLPP-like protein
Peptidase	15	abhydrolase domain-containing protein 2, aminopeptidase N & Q, ATP-dependent Clp protease proteolytic subunit, ATP-dependent zinc metalloprotease YME1, carboxypeptidase M, cathepsin B, cytosol aminopeptidase-like, legumain-lik, probable cytosolic oligopeptidase A, proteasome subunit alpha type-1-like, trypsin-23
Kinase	6	casein kinase II subunit alpha, ceramide kinase, cyclin-dependent-like kinase, serine/threonine-protein kinase LATS1 & PLK4, slowpoke-binding protein
Transferase	8	beta-enolase, creatine kinase M-type, fatty acid synthase-like, phosphoglycerate mutase 1 & 2, protein prenyltransferase alpha subunit repeat-containing protein 1, spermine synthase-like, UDP-glucose:glycoprotein glucosyltransferase
Lyase and Ligase	2	cytoplasmic aconitate hydratase-like, succinate--CoA ligase subunit alpha
ATPase	7	ATP synthase subunit alpha & beta, calcium-transporting ATPase sarcoplasmic/endoplasmic reticulum type, phospholipid-transporting ATPase IIB, sodium/potassium-transporting ATPase subunit alpha, V-type ATP synthase subunit E
Calcium-binding protein	7	calmodulin-like, calmodulin-like protein 4, leucine-rich repeat-containing protein 74B, liprin-beta-1-like, protein eyes shut, ras and EF-hand domain-containing protein, trichohyalin
Cytoskeleton proteins	29	actin, actin cytoskeleton-regulatory complex protein pan1, actin-binding protein IPP, dystrophin, macoilin, myosin-1, myosin-9-like, myosin-IIIa-like, PDZ and LIM domain protein Zasp, protein flightless-1, spectrin beta chain, supervillin, talin-1, tropomyosin 1, twinfilin, unconventional myosin-IXa-like, utrophin-like, keratin II, chromosome-associated kinesin KIF4A-like, katanin p60 ATPase-containing subunit A, kinesin light chain, kinesin-like protein Klp10A, Tubulin alpha-1 chain
Genetic information processing	81	See Appendix A
Heat-shock protein	12	HSP60, HSP70, HSP100 clpX-like
Transporter	14	apolipoprotein D, ABCF3, ABCG4, erlin-1-like, major facilitator superfamily domain-containing protein 6, multidrug resistance-associated protein, nose resistant to fluoxetine protein 6-like, pleckstrin homology domain-containing family G member 3, protein msta, vesicle transport protein SFT2B, potassium voltage-gated channel subfamily H member 2, sodium channel protein Nach-like, Insulin-like peptide receptor, odorant-binding protein
Others and Unknown	41	Insulin-like growth factor-binding protein complex acid labile chain, Band 4.1-like protein 4, peptidoglycan-recognition protein LC-like, huntingtin-interacting protein K, decorin precursor, transmembrane protein 192, etc.

**Table 3 insects-11-00582-t003:** Identified *Oryza* proteins in the BPH honeydew (matched peptides > 1).

UniProtKB ID	Protein Name	Mascot Score	No. of Peptides
A0A0E0H2Z7	glutaredoxin-C6	323	5
Q0D840	thioredoxin H1	195	6
A0A0E0IB24	pathogen-related protein 10	162	4
A0A0E0GKX8	L-ascorbate peroxidase 1	157	4
A0A0E0H8N7	glutathione S-transferase DHAR2	110	3
A0A0E0I7J7	putative acyl-CoA-binding protein RPP10	107	4
A0A0E0ILM1	heat-shock 70 kDa protein	72	2
Q6L4 × 5	thioredoxin H2	60	2
A0A0E0I5B9	Cu/Zn superoxide dismutase	56	2

**Table 4 insects-11-00582-t004:** Identified *Oryza* proteins in the GRH honeydew (matched peptides > 1).

UniProtKB ID	Protein Name	Mascot Score	No. of Peptides
A0A0E0IB24	pathogen-related protein 10	91	2
A6N0C7	polyubiquitin 11	78	2
P0C539	actin-2	50	2
A0A0E0GPY4	disease resistance RPP13-like protein	38	2

**Table 5 insects-11-00582-t005:** Identified bacteria proteins from the BPH honeydew (matched peptides > 1).

Classification	No.	Protein Names *
Oxidoreductases	51	6PGD, ADHE, AHPC, AK2H, ALDH, BETB, BFR, CATA, DHAS, DHE4, DSTOR, ETFD, FABI, FABV, FADB, FRDA, G3P, G6PD, GCSP, GLTB, GPDA, IDH, ILVC, IMDH, ISPG, MAO1, MAO2, MDH, MQO, MSRB, ODP1, PNTA, PNTB, SODF, SODM, TPX
Hydrolases	12	AMY1, APAH, CHIA, DAPE, DEF, F16PA, GCH1, MTNN, SAHH, SUHB, TDCF
Peptidases and Peptidase inhibitors	18	AMPA, CLPP, CLPX, DEGPL, FTSH, GLMS, GUAA, HTPX, LEXA, LON, NPRE, PEPB, PEPT, PIP, ECOT, HFLK
Kinase	8	ARGB, KCY, KGUA, KPYK, PSRP, YEAG
Transferases	81	AAT, ACCA, ACCD, ACKA, ALAA, ARGJ, ARNB, BIOB, CATF, CISY, CYSN, DAPD, DAT, DLDH, DPO1, FABB, FADA, GCST, GLN1B, GLND, GLYA, HIS1, HPRT, ILVB, ILVE, ILVH, ILVI, ISPDF, KAD, KDSA, KPRS, LEU1, LEU3, LEUC, LEUD, MASZ, METE, METK, MURA, NDK, ODO2, ODP2, PFKA, PFLB, PGK, PPNP, PUR2, PUR9, PUUE, PYRB, PYRE, PYRH, PYRI, RISB, SERC, TAL, TKT, TYRB, UDP, UPP, XGPT
Lyases	24	ACEA, ACNB, ALF, ARLY, AROB, ASPA, CAPP, DAPA, FABA, FUMA, HIS7, ILVD, ILVD1, LUXS, PCKA, PCKG, PUR8, RMLB2, THRC, TRPA
Ligases	19	ACCC, ASNB, ASSY, CARA, CARB, DDLA, LIPM, MURC, MURD, PUR4, PUR5, PUR7, PURA, PYRG, SUCC, SUCD
Isomerases	20	DEOB, FKBA, G6PI, GLMM, GPMA, GSA, HIS4, HLDD, PGM, PPIA, RPIA, SLYD, SURA, TIG, TPIS
Translocases	11	CYDA, NUOA, NUOB, NUOCD, NUOH, NUOI, SECA, SECB
ATPases	13	ATPA, ATPB, ATPD, ATPE, ATPF, ATPG, ATP6, ATPL
Ion-binding proteins	2	FETP, NFUA
ABC transporter	10	FLIY, LIVJ, LOLD, LPTB, METQ, OPPD, PSTB, ARTJ, GLNH, GLTI
Other transporters	19	ACP, BRNQ, FTSY, LPP, LPTE, OMP38, OMPA, OMPC, OMPF, PAL, PLPC, PROP, PT1, PTGA, PTHP, PTNAB, TOLB, USPA, YAJC, Y1877
Heat-shock protein	13	CH10, CH60, CLPB, DNAJ, DNAK, GRPE, HSLU, HTPG
Genetic information processing		See Appendix A
Others	7	ARNA, FLIC, FLIG, GCSH, P47K, SP5G1, TYPA
Unknown	10	ERPA, GRCA, SKP, Y1670, Y1887, Y1925, Y2917, Y3794, Y503

* The abbreviate protein names listed in this table are based on the UniProt database.

**Table 6 insects-11-00582-t006:** Identified bacteria proteins from the GRH honeydew (matched peptides > 1).

Classification	No.	Protein Names *
Oxidoreductases	37	6PGD, ADHE, BFR, FABI2, FADB, G3P1, G6PD, GLNA, GLTB, GPDA, IDH, ILVC, IMDH, ISPG, KATG, LEU3, LLDD, MAO1, MAO2, MDH, MQO, MQO1, NUOH, ODP1, PNTA, SODM, THIO, YDFG
Hydrolases	6	ALLC, F16PA, PRZN, SAHH, UPPP3
Peptidases and Peptidase inhibitors	8	CLPP, CLPP2, CLPX, FTSH, HFLK, HTPX, LON, PEPE
Kinases	5	ACKA, ARGB, KAD, KPYK1, KPYK2
Transferases	61	ACCA, ACCD, AK, ARGJ, BCCP, CISY, CYSD, CYSK, DAPD, DAT, DHGA, DLDH, FABB, FADA, GATA, GATB, GLGB, GLMS, GLPK, GLYA, HIS8, ILVD, ILVE, ILVH, ISPB, K6PF, KDSB, MASZ, MEND, META, METE, METK, NDK, PDP, PFLB, PGK, PNP, PPK, PROB, PTGA, PUR7, PUR9, PUUE, PYRH, RISB, SERC, SPEE, TALB, THIC, TYRB, Y2422
Lyases	15	ACEA, ACON1, ACON2, ALF, ARLY, CAPP, CITD, FUMC, PCKG, PRPB, PUR8, RHAD, SPEA
Isomerases	43	ALDH, ASSY, CARA, CARB, FMT, FTHS, GUAA, PUR2, PUR5, PURA, PYRG, SUCC, SUCD, SYA, SYD, SYE, SYFA, SYFB, SYK, SYK2, SYL, SYP, SYR, SYT, SYV, THRC
Ligases	8	FKBA, FKBB, G6PI, GLMM, GPMA, GPMI, TIG
Translocases	4	SECA, SECB
ATPases	11	ATP6, ATPA, ATPA1, ATPB, ATPB1, ATPD, ATPF, ATPG, ATPL
Ion-binding proteins	1	NFUA
ABC transporter	9	ARTJ, DGAL, DPPA, FLIY, GLNH, GLTI, HISJ, LIVB1, METQ
Other transporters	13	ACP, LPP, MSCL, OMP38, OMPA, OMPF, OMPX, PAL, PTGCB, TOLB, YAET
Heat-shock protein	12	CH10, CH60, CLPB, DNAJ, DNAK, GRPE, HTPG, P47K
Genetic information processing	154	See Appendix A
Others	14	FLIC, FLICA, FLAC, FLGG, FLIF, ENGA, ERA, LEPA, TYPA, OPGG, NIFU, ELAB, RCEH, VSP8
Unknown	6	GRCA, LIPM, Y2917, Y356, Y5602, YEAG,

* The abbreviate protein names listed in this table are based on the UniProt database.

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
