# Peer review of "Proteomics of the Honeydew from the Brown Planthopper and Green Rice Leafhopper Reveal They Are Rich in Proteins from Insects, Rice Plant and Bacteria"

_insects, 2020, doi:10.3390/insects11090582_

Round 1
Reviewer 1 Report
Insects Editorial Office
Comments to the Author
From the previous submission, it seems the authors have made significant improvement with the current submission by approaching the diversity of proteins in the honeydew of two plant hopper pests, BPH and GRH by way of proteomic analysis. The proteins identified from two different plant hoppers with different origins (insect host, plants and the symbionts they host) in this study I am sure will generate great interest from applied entomologist, crop protectionists, chemical ecologist and the likes of researchers working on the omics of major pests such as plant hoppers and sap sucking pests that produces honeydew eg. aphids, millie bugs and whiteflies.
The science behind the study has been greatly improved however, I have some minor concerns of which I am sure, are straight forward to solve.
- P1 Line 27; not sure if the term “Unexpectedly” is fitting in this context as honeydew is rich in nutrients that can support any form of life i.e. micro-organisms. I suggest “Coincidently” would be a better term.
- P1 Line 33-34; arrange the keywords in alphabetical order.
- P2 Line 60; replace the phrase “...their harbored microorganisms...” with “the microorganisms they harbor.”
- P2 Line 60; delete “environment”
- P2 Line 62 – 67; This part can be used in the discussion section. Delete.
- P2 Line 69; replace the phrase “this study” with “in which the authors have...”
- P2 Line 69; delete “aphid” as A. pisum is already used. Repetitive.
- P2 Line 72, replace “shortgun LC-MS/MS analyses were” with “...shortgun LC-MS/MS analysis was”
- P4 Line 153; “...midgut, hindgut and Malpighian tubules...” no description in the methods section how the authors digested and extracted proteins from the midgut, hindgut and Malpighian tubules. Please describe clearly how you have digested proteins from these tissues/organs.
- P4 Line 154 and P14 Line 482-483; the citation “[18]”, you CAN only cite works that has been peer reviewed and published. Any works that is still in the preparation or in a peer review process can ONLY be cited as unpublished data. Please remove this citation and cite it as “unpublished data”
- P4 Line 157-158; where is the data confirming that proteins are first secreted intro rice, then ingested into insect guts and then excreted? Please delete this part. I suggest the authors concentrate on proteins that where identified rather than making unsubstantiated claims.
- P9 Line 261; Acyrthosiphon pisum is already mentioned, “A. pisum” is just fine.
- P4 Line 299-300; Please read carefully the article by Yu et al 2016 (10.1104/pp.16.01261).
- P10 Line 307-308. The whole sentence is vague. Replace with “When plant proteins from insect gut are deposited on leaf surfaces, plants are assumed to induce ‘self-recognition’ as a means for defense mechanism.
- P10 Line 309; replace “of” with “in”
- P12 Line 405; replace “of this study” with “in this study”
- P12 Line 406; replace “of rice plant hoppers” with “from rice plant hoppers”
- P12 Line 407; replace “of plant defense” with “for plant defense”
- P12 Line 413; I am not sure what the authors are trying to convey when they say “external environmental microorganisms”!! I understand the message behind but the wording is strange.
- P12 Line 416-417, the phrase “Complex fluid cocktails” is very strange. Please reword or rephrase this part as well. I don’t understand what the authors are trying to convey here.
Overall, the manuscript has been greatly improved, particularly the data and the story in support of the data. I understand the authors have sourced the editing services of an English proof reading company, however, I still feel the manuscript still needs to be improved in its English usage particularly the use of “particles”. I have pointed out some however, there are still major checks that needs to be done by a professional English translator. As such, I STRONGLY suggest the authors seek for a Professional English translator to revise the manuscript (The English-revision should be done NOT by your colleagues but by a professional English editing service). Then, in your following revision (within your point by point list) you must clearly indicate the corrections made by the professional English revisor and provide some of his/her information (e.g. name and email) and acknowledge him/her in the Acknowledgements section for his/her English editing service. If this recommendation cannot be followed, acceptation of your manuscript can be compromised. PLEASE consider this point very carefully.
Author Response
From the previous submission, it seems the authors have made significant improvement with the current submission by approaching the diversity of proteins in the honeydew of two plant hopper pests, BPH and GRH by way of proteomic analysis. The proteins identified from two different plant hoppers with different origins (insect host, plants and the symbionts they host) in this study I am sure will generate great interest from applied entomologist, crop protectionists, chemical ecologist and the likes of researchers working on the omics of major pests such as plant hoppers and sap sucking pests that produces honeydew eg. aphids, millie bugs and whiteflies.
Response: greatly appreciated for the comments and suggestions.
The science behind the study has been greatly improved however, I have some minor concerns of which I am sure, are straight forward to solve.
P1 Line 27; not sure if the term “Unexpectedly” is fitting in this context as honeydew is rich in nutrients that can support any form of life i.e. micro-organisms. I suggest “Coincidently” would be a better term.
Response: according to the suggestion, “Unexpectedly” was replaced with “Coincidently”.
P1 Line 33-34; arrange the keywords in alphabetical order.
Response: arranged the keywords according to the suggestion.
P2 Line 60; replace the phrase “...their harbored microorganisms...” with “the microorganisms they harbor.”
Response: “their harbored microorganisms” was replaced with “the microorganisms they harbor.”
P2 Line 60; delete “environment”
Response: deleted “environmental” according to the suggestion.
P2 Line 62 – 67; This part can be used in the discussion section. Delete.
Response: this part was deleted.
P2 Line 69; replace the phrase “this study” with “in which the authors have...”
Response: “this study” was replaced with “in which the authors have...”
P2 Line 69; delete “aphid” as A. pisum is already used. Repetitive.
Response: “aphid” was replaced with “insect”.
P2 Line 72, replace “shortgun LC-MS/MS analyses were” with “...shortgun LC-MS/MS analysis was”
Response: “shortgun LC-MS/MS analyses were” was replaced with “shortgun LC-MS/MS analysis was”
P4 Line 153; “...midgut, hindgut and Malpighian tubules...” no description in the methods section how the authors digested and extracted proteins from the midgut, hindgut and Malpighian tubules. Please describe clearly how you have digested proteins from these tissues/organs.
Response: this is our custom transcriptomic data which we haven’t published. So there was no description in the methods in the present study. We had not digested proteins from these tissues/organs. We just extracted RNA from these tissues/organs for RNA-seq analysis.
P4 Line 154 and P14 Line 482-483; the citation “[18]”, you CAN only cite works that has been peer reviewed and published. Any works that is still in the preparation or in a peer review process can ONLY be cited as unpublished data. Please remove this citation and cite it as “unpublished data”
Response: according to the suggestion, the citation “[18]” was replaced with “unpublished data”.
P4 Line 157-158; where is the data confirming that proteins are first secreted intro rice, then ingested into insect guts and then excreted? Please delete this part. I suggest the authors concentrate on proteins that where identified rather than making unsubstantiated claims.
Response: this sentence “The presence of salivary gland-specific proteins in the BPH honeydew indicated the possibility that these proteins are first secreted into rice, then ingested into insect guts and then excreted.” was deleted.
P9 Line 261; Acyrthosiphon pisum is already mentioned, “A. pisum” is just fine.
Response: changed according to the suggestion.
P4 Line 299-300; Please read carefully the article by Yu et al 2016 (10.1104/pp.16.01261).
Response: thank you for your suggestion. After reading carefully the article by Yu et al 2016, I found that there is a Ca+-binding protein in rice, OsDEX1, affecting callose degradation. In the previous manuscript, I want to mention that the calcium-binding protein from planthoppers inducing callose deposition in rice has not been determined. However, it is better to delete this sentence. So the sentence “Nevertheless, the calcium-binding protein inducing callose deposition in rice has not been determined.” was deleted.
P10 Line 307-308. The whole sentence is vague. Replace with “When plant proteins from insect gut are deposited on leaf surfaces, plants are assumed to induce ‘self-recognition’ as a means for defense mechanism.
Response: great suggestion. The sentence “We speculated that these rice plant proteins in the BPH and GRH honeydews when deposited on leaf surface of the rice plant may induce ‘self-recognition’ of plant defenses” was replaced with “When plant proteins from insect gut are deposited on leaf surfaces, plants are assumed to induce ‘self-recognition’ as a means for defense mechanism”.
P10 Line 309; replace “of” with “in”
Response: changed according to the suggestion.
P12 Line 405; replace “of this study” with “in this study”
Response: changed according to the suggestion.
P12 Line 406; replace “of rice plant hoppers” with “from rice plant hoppers”
Response: changed according to the suggestion.
P12 Line 407; replace “of plant defense” with “for plant defense”
Response: changed according to the suggestion.
P12 Line 413; I am not sure what the authors are trying to convey when they say “external environmental microorganisms”!! I understand the message behind but the wording is strange.
Response: “external environmental microorganisms” was replaced with “from plant surfaces”.
P12 Line 416-417, the phrase “Complex fluid cocktails” is very strange. Please reword or rephrase this part as well. I don’t understand what the authors are trying to convey here.
Response: “Complex fluid cocktails” was replaced with “Insect honeydew with abundant proteins from different origins”.
Overall, the manuscript has been greatly improved, particularly the data and the story in support of the data. I understand the authors have sourced the editing services of an English proof reading company, however, I still feel the manuscript still needs to be improved in its English usage particularly the use of “particles”. I have pointed out some however, there are still major checks that needs to be done by a professional English translator. As such, I STRONGLY suggest the authors seek for a Professional English translator to revise the manuscript (The English-revision should be done NOT by your colleagues but by a professional English editing service). Then, in your following revision (within your point by point list) you must clearly indicate the corrections made by the professional English revisor and provide some of his/her information (e.g. name and email) and acknowledge him/her in the Acknowledgements section for his/her English editing service. If this recommendation cannot be followed, acceptation of your manuscript can be compromised. PLEASE consider this point very carefully.
Response: The DBediting company improved the revised manuscript and was acknowledged in the Acknowledgement section for the English editing service.
The revisions were listed as below:
L20-21: “Results revealed that the honeydew contains some insect proteins that originate from the saliva and guts, includes some plant proteins that originate from plant sap via the insect digestion system, and also harbors many bacterial proteins from insect or plant symbionts and from plant surfaces” was changed to “Results revealed that the honeydew contains some insect proteins that originate from the saliva and guts, proteins that originate from plant sap via the insect digestion system, and also many bacterial proteins from insect or plant symbionts and from plant surfaces.”
L33-35: “These included saliva proteins, which may have similar functions as saliva proteins involved in rice plantdefenses, such as calcium binding proteins and apolipophorin” was changed to “These included saliva proteins which may have similar functions as the saliva proteins, such as calcium binding proteins and apolipophorin, involved in rice plant defenses”.
L60: deleted “the” before “predators”.
L60: added “the” before “mirid bugs”.
L74-76: “ It also contains symbionts and external environment microorganisms, when deposited onto plant or other surfaces as an excellent growth medium for microorganisms” to “It also contains symbionts and external environment microorganisms. When deposited onto plant or other surfaces, it serves as an excellent growth medium for microorganisms”.
L80-84: “To date, there has been only one publication describing a proteomic analysis of insect honeydew; in which the authors have observed that the pea aphid (Acyrthosiphon pisum) honeydew contained many insect proteins and microbial proteins…” was changed to “To date, there has been only one publication describing a proteomic analysis of insect honeydew from the pea aphid (Acyrthosiphon pisum). The authors reported that the honeydew contained many insect proteins and microbial proteins”.
L161: deleted “the” before “30 adults”.
L295: replaced “yields” to “yielded”.
L351: replaced “were” to “have been”.
L356: changed “both saliva proteomics and honeydew proteomics” to “both saliva and honeydew proteomes”.
L362: changed “both saliva proteomics and honeydew proteomics” to “both saliva and honeydew proteomes”.
L509: deleted “expression” after “overexpression”.
L689: replaced “combination” to “combined”.
L691-692: changed “However, further research is warranted to determine protein quantifications in the honeydew, the functions and roles of the abundant honeydew proteins originating from insects, plants and bacteria” to “However, further research is warranted to determine their quantities in the honeydew, and their specific functions in these interactions”.
Reviewer 2 Report
Comments to the authors
The manuscript by Zhu et al. “Proteomics of the honeydews from the brown planthopper and green rice leafhopper reveal they are rich in proteins from the insects, rice plant and bacteria” identified the proteins present in the brown planthopper (Nilaparvata lugens, BPH) and the green rice leafhopper (Nephotettix cincticeps, GRH) honeydews using shortgun LC-MS/MS analyses. They identified 277 and 210 proteins annotated to insect, 52 and 32 Oryza proteins, and 570 and 494 bacteria proteins in the BPH and GRH honeydews, respectively. They argued that these abundant proteins from insects, plants and microbes may be involved in the multitrophic interactions of plant-insect-microbes. The paper shows an interesting research approach. The study was well-conceived and was very thorough and well done. Although I appreciate the efforts done by authors in conducting such interesting work, I have a couple of minor concerns / made suggestions for improvement that the authors should consider before the acceptance for publication in Insects.
Main concerns:
1- One main concern is that why the authors only identified the honeydew proteins without any quantifications to their amounts in the honey dew. I believe the reader would be very interested in this piece of information. Please provide an explanation for this point in the discussion part.
2- The result section should only include the findings without any interpretations, suggestions, or comparison with previous studies. The latter should be moved to the discussion part.
Specific points are listed below as follows:
Abstract
Line 18: specify which insects.
Keywords
Line 35: add the keyword “protein”.
Introduction
Line 39: provide supporting reference.
Line 51: replace these references (5-10) with more the following references:
- https://pubmed.ncbi.nlm.nih.gov/22559264/
- https://www.ncbi.nlm.nih.gov/pmc/articles/PMC3346861/
- https://pubmed.ncbi.nlm.nih.gov/26123394/
Line 57: add these references:
- https://www.nature.com/articles/ncomms1347?page=2
- https://besjournals.onlinelibrary.wiley.com/doi/full/10.1111/1365-2435.13503
Materials and methods
Line 81: provide a sentence refers to frequent field collections of BPH and GRH to enhance the genetic diversity of your rearing.
Line 103: provide supporting reference.
Results
Line 154: remove this citation from the results section and refer to any papers in preparation as unpublished work with the initials of the first author.
Line 193-196: move these lines to the discussion section.
Line 204-205: move these lines to the discussion section.
Line 247-254: move these lines to the discussion section.
Discussion
Lines 259-263: delete these lines and start with “In the present study”.
Line 312: provide the full name at the sentence beginning.
Line 371: provide supporting reference.
Line 403: provide supporting reference.
References
Line 467: remove “London”.
Line 468: remove “Chapter 122”.
Line 482-483: remove this reference.
Author Response
The manuscript by Zhu et al. “Proteomics of the honeydews from the brown planthopper and green rice leafhopper reveal they are rich in proteins from the insects, rice plant and bacteria” identified the proteins present in the brown planthopper (Nilaparvata lugens, BPH) and the green rice leafhopper (Nephotettix cincticeps, GRH) honeydews using shortgun LC-MS/MS analyses. They identified 277 and 210 proteins annotated to insect, 52 and 32 Oryza proteins, and 570 and 494 bacteria proteins in the BPH and GRH honeydews, respectively. They argued that these abundant proteins from insects, plants and microbes may be involved in the multitrophic interactions of plant-insect-microbes. The paper shows an interesting research approach. The study was well-conceived and was very thorough and well done. Although I appreciate the efforts done by authors in conducting such interesting work, I have a couple of minor concerns / made suggestions for improvement that the authors should consider before the acceptance for publication in Insects.
Main concerns:
1- One main concern is that why the authors only identified the honeydew proteins without any quantifications to their amounts in the honey dew. I believe the reader would be very interested in this piece of information. Please provide an explanation for this point in the discussion part.
Response: thank you for your great suggestion. Actually, Shotgun LC-MS/MS analysis can provide the Exponentially Modified Protein Abundance Index (emPAI) for the estimation of protein abundance. We are planning to conduct further proteomic analysis of the honeydew after different treatments, as previously suggested by one reviewer. In the next manuscript, we will provide the quantitative analysis of the honeydew proteins, which could be more interested by the reader. In the discussion part, I re-wrote the last sentence two sentences: “Insect honeydew with abundant proteins from different origins cast a new light on understanding the multitrophic interaction of plant-insect-microbe. However, further research is warranted to determine their quantities in the honeydew, and their specific functions in their interactions”.
2- The result section should only include the findings without any interpretations, suggestions, or comparison with previous studies. The latter should be moved to the discussion part.
Response: thank you for your suggestion.
Specific points are listed below as follows:
Abstract
Line 18: specify which insects.
Response: “other insects” was replaced with “beneficial insects”
Keywords
Line 35: add the keyword “protein”.
Response: added according to the suggestion.
Introduction
Line 39: provide supporting reference.
Response: added according to the suggestion.
Line 51: replace these references (5-10) with more the following references:
https://pubmed.ncbi.nlm.nih.gov/22559264/
https://www.ncbi.nlm.nih.gov/pmc/articles/PMC3346861/
https://pubmed.ncbi.nlm.nih.gov/26123394/
Response: these three references were added.
Line 57: add these references:
https://www.nature.com/articles/ncomms1347?page=2
https://besjournals.onlinelibrary.wiley.com/doi/full/10.1111/1365-2435.13503
Response: added according to the suggestion.
Materials and methods
Line 81: provide a sentence refers to frequent field collections of BPH and GRH to enhance the genetic diversity of your rearing.
Response: the sentence “Insects were frequently collected from field to enhance the genetic diversity.” was added.
Line 103: provide supporting reference.
Response: The reference (Chen et al. 2007) was added.
Results
Line 154: remove this citation from the results section and refer to any papers in preparation as unpublished work with the initials of the first author.
Response: this citation was replaced with “He et al. unpublished data”.
Line 193-196: move these lines to the discussion section.
Response: moved according to the suggestion.
Line 204-205: move these lines to the discussion section.
Response: moved according to the suggestion.
Line 247-254: move these lines to the discussion section.
Response: moved according to the suggestion.
Discussion
Lines 259-263: delete these lines and start with “In the present study”.
Response: deleted according to the suggestion.
Line 312: provide the full name at the sentence beginning.
Response: “B. tabaci” was changed to “Whitefly”.
Line 371: provide supporting reference.
Response: four references [66, 72, 74, 76] were added.
Line 403: provide supporting reference.
Response: two references (Bao et al. 2012, 2013) were added.
References
Line 467: remove “London”.
Response: removed according to the suggestion.
Line 468: remove “Chapter 122”.
Response: removed according to the suggestion.
Line 482-483: remove this reference.
Response: removed according to the suggestion.
This manuscript is a resubmission of an earlier submission. The following is a list of the peer review reports and author responses from that submission.
Round 1
Reviewer 1 Report
Overall the study was carried out soundly. However, the authors hypothesize about the roles of these proteins in plant immunity. I wish they would carry out an experiment to test this hypothesis.
Reviewer 2 Report
Insects Editorial Office
Comments to the Author
The authors addressed an interesting question and identified a variety of immune related proteins in brown planthopper honeydew, sourcing their origin from three different pedigrees, the plants, the host itself BPH and the bacteria that inhabits in the honeydew perhaps exuded from the BPH gut. The authors also discussed (mainly from literature search) the role of these proteins in Plant-Insect (-microbe) interactions. Their results showed a mass number of proteins identified using shotgun LC-MS/MS proteomic analysis that could play a role in betraying the insects host itself or weakening the plant for insect attack.
Overall, the exact role of these proteins between BPH and rice plants I assume still remains for further studies. The authors present an in-depth study by approaching their question however from one perspective only (shotgun LC-MS/MS proteomic analysis) to identify a mass number of proteins from three different origins. I consider that the study has scientific merits, which make it attractive for publication in the Insects journal. The study presents novel aspects however, the data generated is only from one perspective and thus would need more data from different perspectives and in its present stage, the manuscript needs a strong effort to improve its redaction, especially in leveling up the written English to fully express and portray the study itself.
I have just a couple of concerns which should be addressed:
Major comments
The ‘Introduction’ seems to be all over the place. The authors did a good job by covering every major issue involved in BPH and its honeydew however, the introduction would have been more interesting if it was written in a more methodical fashion. For instance, since BPH is the main host to which the honeydew is exuded i.e. to which a mass number of protein where identified, BPH should have been introduced first, then followed by the honeydew and the proteins that it contains and how the proteins could potentially impact the lifestyle of BPH then state finally state their objectives.
As stated earlier, the study is interesting but addressing the question from one perspective can be shallow i.e. shotgun LC-MS/MS proteomic analysis data only. It is essential to know if the proteins present in the honeydew can really have a major impact on the lifestyle of BPH and basically the interactions between BPH, rice plants and the microbes in the honeydew. For instance, the authors could have perhaps compared the protein signatures between raw honeydew and filtered honeydew. I say this because authors discussed in detail how proteins in the honeydew were of microbe origin and cited literatures that these proteins could be potential elicitors. The Authors also mentioned MAMPs, and MAMPs in my knowledge are very conserved molecules, hence the MAMPs have to be released from the host symbiont (microbe) to be considered as potential elicitors. A simple filtering using different sizes of filter could have directed them to the source of the microbe and its size. Or a simple comparison with heated honeydew to see if the proteins identified are heat stable. In other words, something to show that these proteins found in the BPH honeydew has some biological relevance.
Minor comments
Page 1, line 10-11; the first sentence of the abstract is vague, please rewrite. Page 1, line 11; change ‘unidentified’ to ‘identified’ Page 1, line 20, I think the word plan is missing a letter ‘t’ The last sentence of the abstract, this is merely a future prospective. Since your title reads ‘Brown plathopper honeydew contains plant immune-related proteins of different origins identified by shotgun LC-MS/MS proteomic analysis’, state a conclusion that could substantiate your title. Page 1, line 36; change ‘It was assumed that...’ to ‘It is assumed that...’ since the literature you have stated is still current knowledge. Page 1, line 42; add ‘of’ in between ‘consist’ and ‘waste’ Page 2, line 44; two verbs used in this line, use only one, either ‘contains’ or ‘harbor’ Page 2, line 55; delete ‘still’ Page 2, lines 56-60; the authors describes a publication but there is no reference at the end that states the publication. Page 3, line 138; reorder ‘...cutoff of E-value...’ to ‘...cutoff E-value of...’ Page 4, line 154; two commas at the end of the sentence, remove one Page 4, line 157; add ‘found’ between ‘were’ and ‘only’ Page 4, line 159; I don’t understand what the authors are trying to get across in the sentence that ends with ‘...into rice but ingested and excreted’. Please rephrase. Page 9, line 234; add ‘as’ between ‘(GenBank: AOM6327.1)’ and ‘reported’. Page 10, line 261; perhaps spell out H2O2 as hydrogen peroxide and then later use the formula? Page 11 is very exhausting to read. Paragraph in lines 274-277 is a repetition to the previous paragraph in line 270-273. Since there are so many small paragraphs in this page, can the authors rewrite the paragraph and divide the proteins into the host in which they occur (microbes) or the plants which use to identify the presence of potential herbivores. Or better still, identify them if they occur as elicitors or effectors. If possible, do the same with the paragraphs in page 12. Page 12, line 322; I think the word plan is missing a letter ‘t’ Page 12, line 353; is defoxification a word? Did the authors meant to say, detoxification?
Since I consider that all those redaction problems are solvable, my suggestion is to ask authors that a Professional English Translator review their manuscript before a new submission. Surely, such a reviewing will solve most of those problems and will enable a deeper and more productive referees' reviewing in the next step.